

**Isovolumetric replacement and aeolian deposition contributed to Terrae calcis**
**genesis in Franconia (central Germany)**
Bernhard Lucke[1], Helga Kemnitz[2], Stephan Vitzethum[1]
[1]Institute of Geography, Friedrich-Alexander University Erlangen-Nürnberg,
Wetterkreuz 15, 91058 Erlangen, Germany
[2] Helga Kemnitz, Katharinenholzstr. 33 B, 14469 Potsdam, Germany
Correspondence to: bernhard.lucke@fau.de
Keywords: Isovolumetric replacement, metasomatism, amorphous clay, illuviation,
authigenic clay neoformation, bedrock residue, soils on limestone, Terrae calcis,
Terra rossa, Terra fusca



**Abstract**
We investigated Terrae calcis on limestone and dolomite in Franconia, as well as the
red fill of deep cracks in the rock (*Karstschlotten*). SEM images of the rock-soil
transition zones supported by EDS found amorphous clays along fissures that could
be products of metasomatic, authigenic clay neoformation within microfossils, calcite,
and dolomite grains, or of replacing deposition of amorphous clays inside the calcite,
probably due to percolating waters (illuviation). In the SEM-images, the replacement
appears as exchange process characterized by substitution of Ca and Mg against Si,
Al, and Fe. There is no crystalline clay deposited within rock fissures, and the
transition between calcareous minerals and amorphous clay is gradual. This and the
presence of Fe let it seem possible that plant roots play a major role for the transport
of elements and neoformation of clays, similar to clay pavements along eucalyptus
roots in Western Australia. In this context, more or less uniform Fe(d/t) ratios
contradicting other weathering indicators could be the result of neoformed
phyllosilicates containing $Fe^{3+}$. Bulk soil and bedrock analyses indicate that the solum
of the investigated Terrae calcis does mainly not represent insoluble bedrock residue.
Dust deposition and bioturbation are evident due to sand grains coming from a loess
surface cover, which buried pre-existing Terrae calcis and contributed to their
substrate, apparently supplying quartz and clay-rich pseudosand aggregates.
**1. Introduction**



Until today, there is no general agreement on the genesis of clay-rich red or brown
soils on hard limestone in Mediterranean and temperate climates. It has even been
suggested that they do not represent true soils but a type of claystone (Merino and
Banerjee, 2008). The soil science community largely follows Kubiëna's (1945) model,
which proposes that they represent true soils characterized by accumulation of clay.
He termed mature profiles 'Terra fusca' (with brown color) and 'Terra rossa' (with red
color). Despite the color difference, these soils are characterized by very similar
properties. Their close relationship is expressed by the summarizing group name
'Terrae calcis' (Kubiëna, 1945), which has been adopted in the German soil
classification guidelines (Ad-hoc AG Boden, 2005). Most other soil classification
systems do not use such specific terms for clay-rich soils on limestone, but they are
applied here as we consider them useful for referring to previous works and
discussing the problems connected with the investigation of these soils. For an
extensive summary of the literature and a discussion of classification issues, in
particular the relation to similar soils in the tropics, see Skowronek (2016). Only a
limited part of the literature on the subject can be summarized here; for more
comprehensive information see also the reviews in Merino and Banerjee (2008),
Trappe (2011), Fedoroff and Courty (2013), and Lucke et al. (2012, 2014).

**1.1 Parent materials**
The debate on the genesis of Terrae calcis starts with the parent material. Leiningen
(1930) and Kubiëna (1945) suggested that they resemble mainly the non-soluble
residue of calcareous rocks after in-situ dissolution of limestone by meteoric water.
However, already these authors contemplated that Terrae calcis may contain a major



allochthonous component, possibly mainly from aeolian dust. As outlined by Schmidt
et al. (2006), it seems well possible that the contributions of sources vary locally.
Studies in Italy (Moresi and Mongelli 1988), China (Shijie et al. 1999; Ji et al. 2004a,
2004b), and Turkey (Temur et al. 2009) support the residual theory since the mineral
assemblies and geochemistry of these soils are largely similar to the non-calcareous
residue of the underlying limestones. However, at other locations substantial
differences between soil and rock residue have been found (e.g. Leiningen 1915;
Durn et al. 1999). As well, it seems questionable at some locations whether residual
formation out of pure limestone could be possible, as this would require the
dissolution of huge amounts of rock (Yaalon and Ganor 1973; Merino and Banerjee
2008). In this context, long-range transport of Saharan dust was found to play a
significant role for soil formation on limestone in a large part of the northern
hemisphere (e.g. Muhs, 2001; Muhs et al., 2007; Lucke et al., 2014).

**1.2 Mechanisms of soil formation: aeolian deposition, bedrock dissolution, and**
**illuviation**
Closely connected with the question of parent materials is the mechanism of soil
formation. Since a steady deposition of Saharan dust takes place in the
Mediterranean (Martin et al. 1989), Yaalon and Ganor (1973) suggested that clay-
and Fe-rich dust settling with precipitation is the main substrate of most Terrae calcis
in that area. Saharan dust could even be traced in Terrae calcis of the West Indies
(Muhs 2001; Muhs et al. 2007; Prognon et al. 2011). Danin et al. (1982) found fossil
marks of lichen on limestone under a Terra rossa in Israel, suggesting that that the
rock had once been exposed to sunlight before being covered by soil. In northern



Jordan, Lucke et al. (2014) found a continuous dust signal in soils on different
bedrocks and concluded that aeolian deposition must have provided a significant
amount of the soil parent material, even though a specific and significant contribution
of each different bedrock was clearly indicated as well. Therefore it seems possible
that a mixture of aeolian deposition and bedrock weathering can contribute to the
genesis of Terrae calcis.

Dissolution of limestone, in particular of clay-rich marls that are often interbedded in
calcareous formations, could directly produce a clayey residue (Bronger et al., 1984).
During assumed long periods of soil formation, such clay-rich interlayers could have
disappeared due to weathering, which is why a divergent composition of the soil and
the now underlying limestone does not necessarily prove an allochthonous origin of
the solum. In this context, Frolking et al. (1983) and Fedoroff and Courty (2013, and
references therein) suggested an illuvial origin of the phyllosilicates. Since smectites
may maintain surface acidity even when dispersed in Ca-rich water (Mortland and
Raman, 1968), clays might be transported by subsurface waters, trigger limestone
dissolution, and accumulate, leading to an effective replacement of limestone by clay.
As chert bands in the Galena dolomite in Wisconsin continued through the clays
studied by Frolking et al. (1983), this replacement must have taken place
isovolumetrically, i.e. not creating larger voids so the chert bands were not disturbed.
That the clays had been transported was indicated by oriented coatings on slightly
weathered dolomite in greater amounts than could result from in-situ rock dissolution
(Frolking et al., 1983).



### 1.3 Metasomatism in Terrae calcis genesis


An alternative way of Terrae calcis genesis was first suggested by Blanck (1915): the
clay-rich substrate could not only be product of bedrock weathering, illuviation, or
aeolian deposition, but consist of newly formed (authigenic) clay minerals that
replaced the limestone in a pressure-driven metasomatic reaction. In this context,
Stephenson (1939) and Ross and Stephenson (1939) discovered fossil mollusks that
had apparently been replaced by beidellite and were preserved as clay structures in
a limestone bed near Pontotoc, Mississippi. Monroe (1986) reviewed literature
regarding the replacement of limestones by clay and conducted a replacement
experiment based on circulation of mineralized groundwater through limestone: the
experiment failed, but the reported evidence suggested that limestone could indeed
be replaced by lime-free clay. These examples, however, do not only refer to the
genesis of soils on limestone, but to clay layers within limestone beds and to clay fills
of subterranean caves (see also Zippe, 1854; Weyl, 1959; and Fenelon, 1976).

Maliva and Siever (1988) simulated in the laboratory how chemically completely
strange guest minerals can grow in host minerals if a superconcentration of ions
precipitating as guest mineral is present: the guest mineral replaces the host mineral
in a pressure-driven reaction while maintaining its bedding structures (by 'force of
crystallization'). In this context, Zhu and Li (2002) described metasomatic relic
bedding structures of the underlying limestone in Terra Rossa in southern China,
which could be result of replacement processes as simulated by Maliva and Siever
(1988). Merino and Banerjee (2008) investigated thin sections of a 'bleached zone' in
the rock-soil transition zone of a Terra rossa in Bloomington, Indiana, and described a



'reaction front' that was characterized by partial isovolumetric replacement of
limestone by clay, as well as dissolution voids that were associated with the
replacement. They calculated a thermodynamic model of the replacement reaction.
According to this model, the reaction would on the one hand lead to a pressure-
driven isovolumetric replacement of the limestone during authigenic clay
neoformation, meaning that the mass balance of soil formation versus bedrock
dissolution requires much less limestone for soil formation than the residue model.
On the other hand, the reaction would produce acids which could explain the
association of Terrae rossa with karst, and lead to additional chemical dissolution of
limestone with a subsequent mixing of the non-soluble residue with the newly formed
clay.

However, even the comprehensive model presented by Merino and Banerjee (2008)
cannot fully explain (authigenic) clay neoformation: one major question is the supply
of ions into the rock fissures. Merino and Banerjee (2008) suggested that dissolved
aerosols would deliver the necessary elements (mainly Si, Al, and Fe). Banerjee and
Merino (2011) further refined the replacement model by accounting for diffusion and
infiltration processes. These were based on Amran and Ganor's (2005) dissolution
half-lives of smectites in water – for pH 5. However, considering the often neutral to
slightly alkaline pH-values as e.g. found by Lucke et al. (2012) for most Terrae calcis
in northern Jordan, it seems still questionable how ions of Al that are hardly soluble
under such conditions could be mobilized from the surface into the rock pores.
Metasomatic features could be relic and have formed when soils were completely
decalcified, which seems possible since there is evidence that Terrae calcis in Jordan



were subject to re-calcification in the recent past (Lucke, 2008). However, organic
matter might play a role too: Blanck (1915, 1926), Blanck et al. (1928) and Blanck
and Oldershausen (1936) proposed that organic acids provide colloids that prevent
ions from precipitating even when in contact with calcareous rocks (see also the
comprehensive review in Blanck (1930). This idea could explain why red soils are
largely absent on calcareous rocks of temperate zones: due to the much lower
humus contents of soils in Mediterranean climates, ions must precipitate when
reaching the calcareous rocks, but not if larger humus concentrations are present
(Blanck, 1915). It could mean that Terra fusca might be less a product of replacement
processes than Terra rossa due to the usually significantly higher contents of organic
matter in Terra fusca.

However, other ion transport mechanisms seem possible too: Reifenberg (1927,
1947) argued that silicic acids, and not humus, provided the colloid that prevented
flocculation of sesquioxides and growth of minerals before ions entered the rock
pores. Reifenberg (1947) suggested further that the source of the ions in semi-arid
areas might not be the soil surface, but ascending waters from the rocks. Lucke et al.
(2012) suggested that plant roots might supply the necessary ions to the rock pores,
based on the observation that plant roots can often be found in rock fissures, and on
the reported neoformation of clays associated with root exudates of mallee eucalypts
in geochemically completely different sand dunes that was observed by Verboom et
al. (2009). This casts some doubts on a possible connection of organic matter and
colors of Terrae calcis, although Terefe et al. (2008) suggested that repeated
vegetation fires and thus organic matter could cause long-term red coloration.




### 1.4 Colors of Terrae calcis

Various ideas have been brought forward to explain the prevalence of brown and red
colors of Terrae calcis. The red color of Terrae rossae could be inherited from the
insoluble limestone residue (Bronger et al., 1984), could have formed during
pedogenensis under warmer and more humid climates of the past (Klinge, 1958), or
simply result from oxidized $Fe^{2+}$ that was released during the weathering of
carbonate rocks (Meyer, 1979). Schwertmann et al. (1982) suggested that rapid
wetting-drying cycles as often prevailing in Mediterranean climates and on well-
drained karst areas can lead to recrystallization of ferrihydrite as hematite, which
gives red color even when present in only small concentrations. Although this
process could not yet be modeled in the laboratory, Barrón and Torrent (2002)
showed that ferrihydrite can transform into maghemite under presence of phosphate
or other ligands capable of exchange of Fe-OH surface groups. Based on this,
Torrent et al. (2006) suggested that maghemite formation is a precursor of hematite
during ferrihydrite transformation in aerobic soils poor in organic matter, which
matches evidence collected by Lucke and Sprafke (2015) along a climatic transect in
northern Jordan. In contrast, the prevailing brown colors of temperate areas seem
connected to less pronounced moisture differences, and it has been suggested that a
change to moister and cooler conditions could have caused xanthization of formerly
red soils, meaning that red-colored hematite would change to goethite (Boero and
Schwertmann, 1987).



### 2. Genesis of Terra calcis in Franconia (central Germany)

Franconia hosts widespread limestone plateaus which are partially covered by brownish and reddish Terra calcis (locally called *Alblehm*). These plateaus were mostly not glaciated, but situated in a periglacial environment during the Pleistocene. Terraes calcis are partly present at the surface, and partly buried by loess and sands, and can be found as fills of deep cracks and dolines in the limestones. While the plateau soils are mostly brown, the crack fills are often characterized by intense red colors. Their environmental significance is debated. According to Mückenhausen et al. (1975), red infillings in limestone cracks could be relics of a former soil cover, possibly of Terrae rossae or Ferralsols from the Cretaceous or Tertiary, which was eroded on the surface but preserved in the cracks. Zech et al. (1979) investigated red clayey fills in karst cracks of Franconia and concluded that their color testifies to formation during warmer and moister tropical climates of the past. This is supported by higher amounts of kaolinite in the clay infillings compared to the residue of the surrounding bedrock. However, in the light of theories of Terrae calcis formation discussed above, it seems also possible that the crack fills are products of illuviation, or clay neoformation – and the different color might be explained by divergent organic matter contents, since the cracks are not connected to the actual surface.

In this context, "powdery" layers of $CaCO_3$ of approximately 1 mm thickness have been described by Trappe (2011: 96) for the rock-soil transition zones of Terrae calcis in our studied area in Franconia, who interpreted them as potential evidence of insoluble limestone residue contributing to solum formation. However, the limestones are very pure: Häusler and Niederbudde (1992) estimated that about 3 mm of soil



cover could have formed out of the respective limestone residue during the
Holocene, meaning that Terrae calcis such as the ones investigated in this study
would have experienced about 2.7 Ma of soil development under the current climate
without erosion in order to reach the present depth. Trappe (2011) argued in this
context that the Terrae calcis and crack fillings in Franconia represent mixed
sediments, largely stemming from weathered chalk layers, since the limestones
produce too little residue. However, no high-resolution study of the micromorphology
of these soils and their transition zones to bedrock was yet accomplished.

Although a final explanation of the mechanisms of the metasomatic replacement
process cannot yet be offered, there is growing evidence that in-situ neoformation of
clay can contribute to the formation of clay-rich soils on limestone (see e.g. Feng et
al. (2009) and the review by Laverty (2012). Key evidence for isovolumetric
replacement are 'shadows' of the original rock structures that are preserved in the
soil-rock transition zone, in particular microfossils partially consisting of clay. If
neoformation of clay minerals in limestone took or takes place, a spatially precise
approach focusing on partially replaced microfossils can track it. And if partially
replaced microfossiles can be confirmed for Terrae fuscae, it would suggest that
metasomatic processes are less dependent on organic matter than suggested by
Blanck (1915). Therefore we studied the micromorphology of the rock-soil transition
zones of two Terrae fuscae and a limestone crack filled with Terra rossa in Franconia
with regard to the presence of partially replaced microfossils and minerals. In
addition, bulk samples of the substrate and the microsurfaces of sand grains were
investigated. These areas had been studied before by Blanck and Oldershausen



(1936), and we attempted to re-visit some of their investigation sites in order to honor
their prior efforts, applying now available more advanced methods of analysis.

**3. The sampled profiles**

From a total of five studied sites, two are presented here. The first profile 'Fricke' is
located in a quarry approximately 4 km southeast of the town Weißenburg i. Bayern
on top of a limestone plateau (N 49° 00' 36.6", E 11° 01' 35.7", see figs. 1, 2a and b).
Here a thick hard limestone from the upper Jurassic/Malm δ containing *Ammonites*
*pseudomutabilis* (locally called *Weißenburger Marmor*) is exposed, which is well-
suited for construction and exported worldwide. The rock has vertical cracks that are
filled with uniform red clay which was already observed and studied by Blanck and
Oldershausen (1936). We named our profile according to the designation given by
these authors, although the exact part of the limestone quarry which they studied has
been removed by constant quarrying during the past 80 years. In the studied profile,
the clay-filled cracks are not connected to the present surface, but interrupted by
another layer of hard limestone. On this layer a zone of intense rock weathering (see
fig. 2b) resembles the bedrock of soil formation on the current surface. It can
however not be excluded that there was a connection of the crack to the surface in
the front of the profile that was removed by quarrying.

The soil developed at the surface can be classified as Cambisol (Siltic) according to
the World Reference Base of Soil Resources (WRB 2014). According to the German
soil classification system (Ad-hoc AG Boden 2005), a Braunerde-Terra fusca is
present: the lower part of the profile could be described as Terra fusca, which



281 gradually changes into a Braunerde formed out of loess in the upper part. Soil

282 horizons were classified according to the German system (Ad-hoc AG Boden 2005,

283 see figs. 2a and b). There is a gradual transition from a dark-brown, clay-rich Terra

284 fusca horizon (II TBv) to a bright yellowish, silty Bv horizon that apparently resembles

285 loess which at some time buried a prior developed Terra fusca. Roots were present

286 throughout the whole profile and even in the deepest crack fillings. Although we

287 sampled the whole profile for bulk soil analysis, only the soil-rock transition zone of

288 the red clay in the cracks was studied by micromorphology. Bulk samples of the crack

289 infillings were taken in the upper and lower part of the crack (samples III Tu (1) and

290 (2). The crack fillings are homogeneous and no horizons or indicators of fluvial

291 deposition could be observed (fig. 2c). As well, the bedrock was sampled and

292 analyzed for calcium carbonate, total element contents, residue particle sizes, and

293 residue color.

294

295

296 Figure 1

297 Figure 2a

298 Figure 2b

299 Figure 2c

300

301 The second profile 'Schwaighauser Forst' is located at the eastern border of

302 Franconia, locally called *Bruchschollenland*, an area characterised by strong faulting

303 and dislocation of geological units. Therefore very different lithologies are exposed at

304 the surface in small areas, and some volcanic activity about 50 km to the east was

305 associated with the faulting. The studied soil represents an Epileptic Cambisol





(clayic) according to the WRB (WRB 2014). According to the German soil
classification system (Ad-hoc AG Boden, 2005) it can be classified as Terra fusca. It
is located in an educational soil trail (N 49° 05' 10.7", E12° 00' 12.7") north-west of
the city of Regensburg (TUM 2014), formed on dolomitic limestone from the Upper
Jurassic/Malm ε-ζ. At this profile, a loess cover is not discernible, but may have
eroded since the solum is much shallower than at the first profile (figs. 3a and 3b).
Roots were present throughout the whole profile. Apart from each horizon, the
bedrock was analyzed for calcium carbonate, total element contents, residue particle
sizes, and residue color.


Figure 3a
Figure 3b

## 4. Methods

Our main aim was to check whether partially replaced microfossils can be found in
the rock-soil transition zones of the sampled profiles, supported by some analyses of
bulk soil and bedrock residue in order to describe soil development intensity.
Micromorphological samples were taken from the transition of the unweathered
bedrock to the clay of Terrae calcis, including several samples covering the whole
distance from the apparently unweathered rock till clay aggregates of the solum. Thin
but stable metal containers were placed on the transition of soil and rock and the
samples slowly cut out with a knife. After freeze drying, samples were stabilized using
Araldite A2020 epoxy resin. The thin sections received a diamond-polishing and were



carbon coated. After optical analysis, microanalysis was done with a high resolution
field-emission scanning electron microscope (FESEM; Carl Zeiss Ultra 55 Plus),
equipped with an energy-dispersive system (EDS) by Thermo Fisher Scientific. In
contrast to only optical analysis, scanning electron microscope analysis with EDS
permits determining the geochemistry of the studied areas. This can allow detecting
minerals that appear amorphous to X-ray diffraction and optical studies. In addition,
large allochthonous clays such as biotite grains can be misinterpreted as in-situ
formed minerals as shown by Lucke et al. (2012) if only optical analysis is applied,
and the scanning electron microscope (SEM) allows investigating smaller features at
scales of nanometers.

General soil analyses determined calcium carbonate and organic carbon contents
with a Leco TrueSpec C-N analyser measuring samples before and after ignition of
organic matter at 430 °C for two hours (Schlichting et al., 1995), based on the
assumption that the remaining content of C represents carbon bound in calcium
carbonate. The dolomite content of the dolomitic limestone was estimated from the
residue mass after dissolution of the calcareous part of the rock with 10% HCl. Soil
color of dry soil samples and the color of dried bedrock residue after dissolution with
10% HCl were determined using the Munsell color chart, and redness ratings
calculated according to Hurst (1977). In order to control whether the treatment of the
rock with 10% hydrochloric acid could lead to a loss of red color of the residue,
samples of the Terra rossa limestone crack infillings were simultaneously treated with
the same amount of acid during the same time of bedrock dissolution, but no color
change could be observed. Analysis of clay minerals would require rock dissolution



with weaker acids in order to exclude alteration of the minerals due to acid treatment
(Ostrom, 1961; Rabenhorst and Wilding, 1984), but since only the texture and color
of the residue were studied, faster treatment with stronger acid was chosen.

Pedogenic iron oxides were extracted with sodium dithionite at room temperature
according to Holmgren (Schlichting et al. 1995), and the iron contents measured with
an ICP Spectrometer (Thermo Scientific iCAP 6200 Duo). In case samples contained
more than 4% $CaCO_3$, particle sizes were analyzed after removing $CaCO_3$ with 10%
hydrochloric acid and washing the samples until conductivity dropped below 200 µS.
These and the other samples containing less than 4% $CaCO_3$ were then dispersed
with sodium hexametaphosphate ($Na_4P_2O_4$) and shaken overnight (Schlichting et al.,
1995). Wet sieving determined the sand fraction according to DIN 19683 (1973),
while the smaller particles were analysed with a Sedigraph (Micromeritics). For total
element analysis by X-ray fluorescence, we determined the loss on ignition (LOI) by
weighing the powdered samples before and after drying: 1) 12 hours at 105 °C in a
cabinet dryer and 2) 12 hours at 1030°C in a muffle furnace. Major element oxides
($SiO_2$, $TiO_2$, $Al_2O_3$, FeO, MnO, MgO, CaO, $Na_2O$, $K_2O$, $P_2O_5$) and selected trace
elements (Ba, Cr, Ga, Nb, Ni, Pb, Rb, Sr, Th, V, Y, Zn, Zr) were measured with a
Spectro XEPOS at the GeoZentrum Nordbayern. Precision and accuracy are
generally better than 0.9% and 5%, respectively.

**5. Results**
**5.1 Micromorphology of the rock-soil transition zones**



Limestone beddings partially consisting of clay could be found in the rock-soil
transition zones of both studied profiles. Figure 4 shows a calcite grain partially
consisting of clay in the 'powdery' (see Trappe, 2011) transition zone of about 1 mm
thickness, directly between the red clay and the limestone in the profile 'Fricke'. It
appears that a prograding solution is encompassed simultaneously by the formation
of clay. As shown by the serrated grain rim in fig. 4 (upper part), exchange seems to
occur along zones of potential permeability such as fine fissures, at grain contacts, or
through pores. That part of the calcite already consists of clay is shown by the
spectral image in fig. 4 (lower part). Although secondary calcite needles were noted
in some fissures, no clay could be observed in the pores: there are no clay films
suggesting that allochthonous clay has been transported into the rock. As well, the
original form of the calcite grain is so well preserved that the clay cannot be of
allochthonous origin, but must represent the in-situ limestone, and there is no
structure discernible that could be attributed to micro-clay beddings deposited during
limestone formation. Conclusively, bedrock weathering may not simply proceed by
chemical reaction processes which create voids in the rock due to chemical
dissolution, but clay neoformation and rock dissolution could be part of the same
process as suggested by Merino and Banerjee (2008).

Figure 4

About 1 cm deeper into the limestone, it appears rather unaltered to the naked eye
and remains of biogenic shells can be observed macroscopically. But there are also





some small reddish lines. Analysis with the SEM confirmed that some of these lines
represent clay, which is partly present inside microfossils. Figure 5 shows a biogenic
relict (probably alga or foram) partially consisting of clay which is surrounded by still
largely unweathered limestone. Again, there is no evidence that the clay was
transported into the microfossil, since the outer shell is still closed and consists of
calcium carbonate. Though the outer shell seems preserved and closed, consisting of
calcium carbonate, one has to account for the three-dimensionality of the organism
and a high likelihood of pores. But the apparently amorphous structure of the clay
inside the microfossil argues against allochthonous clay minerals. Furthermore, it still
contains significant amounts of the former calcium carbonate filling (see spectral
images in fig. 5).

Figure 5

The Terra fusca of the Schwaighauser Forst profile formed on a dolomitic limestone
that is characterized by low density of microfossil remains in the rock. Yet, some
microfossils partially consisting of clay could be observed in the direct soil-rock
transition zone, inside the rock in about 10 µm distance to the border of the limestone
as shown in figure 6. This microfossil is characterized by high contents of magnesium
corresponding to bedrock chemistry. The minor Si peak that is detected in the inner
section might either derive from naturally implemented traces of silica or indicates
authigenic neoformation of clay, too.





Figure 6

In a clay-filled fissure of the dolomite rock beneath the Terra fusca, we investigated
serrated structures partly showing areas that appear darker in the SEM. Figure 7
shows a calcite grain directly bordering the clay-filled fissure. The EDS-analyses
show how the calcite at the edge starting to disintegrate contains already some silica
and magnesium (point 1). Since magnesium might also be a component of the
carbonate, one might argue for an analytical effect because of the beam size. This,
however, does not fall below under 1.5 µm in diametre in this case, such that the
analyzed microvolume is restricted to the calcitic part of the grain. Comparison with
the elemental composition of the clayey crack fill with the clay in the fissure (point 2),
in particular regarding the Al:Si ratios, shows a clear clay mineral signal with a
strongly reduced calcium content.

Figure 7

The replacement of calcium and magnesium by clay seems further indicated by an
EDS cross-section (line scan) through a dolomite grain about 50 µm deeper into the
bedrock (fig. 8). A darker domain of the dolomite proves as clay, characterized by a
significant decline of Ca and Mg, while Si, Al, and Fe increase. However, this clay
seems not yet fully de-calcified, and appears merely like an amorphous gel. We have
not been able to identify crystallinity, and the area is too small to be determined by
optical analysis, which would misinterpret it as calcite.




Figure 8

**5.2 Bulk soil analysis**
Results of both profiles are summarized in table 1. It can be observed that the Terra
rossa red crack filling samples of the Fricke profile [samples Tu(1) and Tu(2)] are
more or less identical, confirming the field impression that no horizons or deposition
patterns are present in the cracks. However, the fills are distinct from the Terrae
fuscae: apart from the red color, values of dithionite soluble iron are elevated. As well,
values of oxalate-soluble silicium are elevated, pointing to a higher content of
amorphous silica than in the Terrae fuscae.

The Terra rossa crack fillings are characterized by higher clay contents than the
Terrae fuscae, but also by a higher sand content than the bedrock residue. The latter
could to some degree be connected with the acid treatment to dissolve the rock. The
crack infillings received no acid pre-treatment before grain size analysis due to low
$CaCO_3$-contents, and since acid-pretreatment also removes calcite sand grains
(Lucke and Schmidt, 2015), sand contents of crack infillings and bedrock residue are
not directly comparable. Calcite sand grains in the Terra rossa-filled cracks might
represent micro-"floaters" as suggested by Meert et al. (2009) regarding limestone
blocks of larger scale "floating" in the Bloomington Terra Rossa in Indiana. However,
the majority of sand grains in the crack fillings consist largely of rounded quartz
grains of the finer sand fractions (6% coarse sand, 47% middle sand, 47% fine sand),

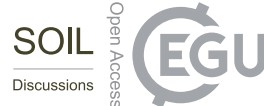

with some equally rounded black grains that might reflect a primary detrital input.
While some quartz grains of the coarse-sand fraction show densely set small V-
shaped marks typical for aeolian transport during loess deposition (fig. 9), there are
also grains being completely covered by clay minerals that are unsuitable for a grain
surface characterization (fig. 10).

Figure 9
Figure 10

This suggests that allochthonous material, including grains transported with the
overlying loess, was involved in the formation of the crack fills. The very similar sand
contents in the two samples of crack fillings point to a very homogeneous distribution
of sand within the cracks, which is in agreement with the field impression and other
results of bulk soil analysis. The strongly elevated sand contents compared to the
bedrock residue argue against inheritance from rock dissolution, and the absence of
fluvial sorting and very homogeneous distribution of sand grains in the clay argue
against fluvial deposition of the crack fills.

Although the crack fillings were affected by loess deposition on the surface, lab
results support the field impression that two different soil formation processes took
place. Very similar Ti/Zr values of the lower Terra fusca at the profile Fricke [sample II
TCv] and the Terra rossa red clay crack fillings let a common parent material seem



possible, which is different from the limestone residue. Although it cannot be ruled out
that clay-rich beds - now removed due to weathering - provided residue contributing
to soil formation, the analyzed rock sample suggests that neither the buried Terra
fusca nor the Terra rossa crack fillings originated from the residue of the now
adjacent limestone.

Table 1

The limestone and dolomitic limestone proved to be very pure with acid-soluble
fractions of 98.1% and 99.99%. Their residue is extremely clay-rich compared to the
crack fillings and the II TCv sample of the Terra fusca of the Schwaighauser Forst
profile. As well, Ti/Zr ratios are different between rocks and soils, although there are
gradients visible in the profiles. These supports the field impression that loess was
deposited on the surface and mixed with pre-existing Terrae calcis, leading to a
gradual lowering of the Ti/Zr ratio towards the tops of the profiles. Even though there
was no loess addition apparent in the Schwaighauser Forst profile, particle sizes
clearly support that silty material was mixed into the soil. This is further stressed by
the absolute content of dithionite-soluble iron, indicating that less weathered material
was deposited on top of the profiles. However, in contrast to the other parameters,
Fe(d/t) ratios suggest that weathering intensity remained nearly constant through the
profiles. This is not an uncommon phenomenon: Günster (1999) and Lucke (2008)
encountered similar Fe(d/t) values in Terrae calcis of southern Spain and northern
Jordan, which did not match other indicators of pedogenesis intensity. It could be



explained assuming that some of the dithionite-extractable iron in well-developed
Terrae calcis might not result from oxidation of $Fe^{2+}$ during weathering, but represents
pre-weathered $Fe^{3+}$-rich phyllosilicates released from the rock residue – or enters the
system from outside as $Fe^{3+}$. The latter might be happening when mineral
neoformation due to isovolumetric replacement, illuviation of amorphous clay, or dust
deposition contribute to soil formation.

The contents of organic matter and calcium carbonate are mostly low throughout the
profiles. The Terrae fuscae seem to contain more organic material than the Terra
rossa of the crack fillings, which matches expectations. However, in the profile Fricke
only the Ah-horizon contains a relatively high amount of organic matter, while the
lower horizons have only little more than the crack fillings. On the one hand, it is
surprising that the crack fillings do contain organic matter at all, since they seemed
disconnected from the present surface. On the other hand, roots occur in limestone
fissures even in great depth, so that the organic matter present in the limestone
cracks most likely represents roots.

pH values are mostly neutral – only in the upper part of the Fricke profile values
between 4.5 – 5.5 can be observed, but this matches the high amount of organic
matter and the field impression that the parent material of this part of the profile is
mainly decalcified loess. There is no difference of the pH values of the red clay
infillings of the limestone cracks and the Terra fusca of Schwaighauser Forst. In this



context, the bedrock residue of both profiles is not red, but brown and greyish brown.
Therefore it seems unlikely that the color was inherited from the bedrock residue.

Last but not least, calcium carbonate values are low throughout all profiles, except in
the TCv horizon of the Terra fusca developed on dolomitic limestone in the
Schwaighauser Forst profile. Apparently the dolomite has not completely been
weathered, and thin sections showed that calcium carbonate re-precipitated in the
soil matrix.

**6. Discussion**
**6.1 Parent materials**
It is evident that the investigated Terrae fuscae and Terra rossa crack fillings cannot
represent only bedrock residue. This is indicated by the differences of color and
particle size between the soils and bedrock residues. In addition, grains of the sand
fraction of the crack fills confirm that there has been a deposition of material from the
covering loess into the fills. Moreover, at the microscale it seems possible that
metasomatic processes took or take place in the direct rock-soil transition zones of all
investigated profiles, leading to replacement of bedrock with clay in the
approximately outer 50 µm of the bedrock. In this context, clay neoformation seems a
gradual replacement process: the SEM observations point to a gradual exchange of
Ca and Mg against Si, Al, and Fe. Newly formed (authigenic) clay appears as gel-like
material, distributed along calcite as well as dolomite grain boundaries, or in patches
and small domains within and around decaying carbonate rock and its incorporated



microfossils. No crystallinity can be identified. Often, the contact between
calcite/dolomite and clay is present as an irregularly prograding, indenting front,
which is important to note as it constrains a simultaneous transformation from one
mineral to the other.

**6.2 Amorphous (metasomatic?) clays**
All clays occurring in microfossils and minerals are connected to grain boundaries,
fissures, and pores occurring within the soil-rock transition zone, while the unaltered
rock in areas remote from fissures lacks those observations. Thus underlines a
correlation between the replacement process and the movement of solutions in the
rock. There are no deposition structures such as oriented crystalline clay layers that
could be connected with illuviation. However, it cannot be ruled out that amorphous
clay is transported into the transition zones. If amorphous phyllosilicates are
transported by percolating waters as proposed by Frolking et al. (1983), they are
apparently not accumulating in voids, but adhere to calcite and dolomite grain
surfaces. Most of the observed phyllosilicates contain iron, implying that it has either
been transported as coating of amorphous clay minerals, or as ions. At few locations
secondary $CaCO_3$ needles precipitated near fissures, suggesting at least temporary
presence of solutes in the pore water. There is no discernible microstructural
difference between Terra fusca and Terra rossa in the scanning electron microscope
images.

**6.3 Bulk soil analysis**



Bulk soil analyses let it seem possible that replacement and/or transport of
amorphous clays contributed more to the genesis of Terra rossa in the crack fillings
since the contents of oxalate-soluble silica are higher in the Terra rossa. However,
there is no reason to assume that higher contents of organic matter in Terrae fuscae
prevent or strongly limit processes of metasomatism and/or transport of amorphous
clays. This argues against Blanck's (1915) suggestion that topsoil organic matter
contents play a role in Terrae calcis genesis. In this context, the very similar pH
values of the Terra rossa crack infillings and the Terrae fuscae suggests that the pH
value does not play a major role for the development of red or brown color – at least
if assuming that the actual pH is relevant for amorphous clays and the formation of
replacement features. Similarly, it does not seem probable that organic matter
contents affect the color of Terrae calcis since the differences between Terra rossa
and Terra fusca are not very pronounced, at least in the Fricke profile.

The Terra rossa and Terrae fuscae are characterized by similar particle sizes. At both
sites the particle sizes indicate deposition of loess in the upper part of the profiles.
Since the bedrock residue is at both profiles characterized by much finer particle
sizes, it seems unlikely that the residue of bedrock dissolution contributed a major
part of the solum. This is further supported by the Ti/Zr ratios, which are different
between the rock and overlying soil. They indicate that loess did not provide the main
parent material of Terrae calcis genesis, but altered soil properties during deposition
on pre-existing soils. There is a strong similarity between the Terra rossa crack fillings
and lower part of the Terra fusca in the Fricke profile, which lets it seem possible that
similar parent material was involved in formation of these soils. Whether this was



aeolian dust cannot be deducted from the available data: the Ti/Zr ratios support the
impression of loess deposition and mixing only into the upper part of the profiles, but
do not deliver insights into possible aeolian parent materials of earlier pedogenesis.
Here the dominant finer sand fractions in the Terra rossa crack fillings, as well as the
absence of horizons or depositional structures connected with fluvial sediments,
support that an early phase of aeolian sedimentation might have been connected
with formation of the red fills. These aerosols could have delivered elements driving
replacement reactions as well. The very homogeneous distribution of sand grains in
the clay of the crack fill speaks against fluvial sorting patterns. The most likely
explanation how sand grains entered the cracks seem root channels and shrink-swell
cracks in the clay, while the dominance of the finer sand fractions indicates an
aeolian source of the grains – possibly largely from loess deposition.

Similar to the sand, small aggregates consisting of silt and clay could have been
transported into the limestone cracks by wind and bioturbation. Fedoroff and Courty
(2013) suggested that wind-blown transport of clayey pseudosands from pre-existing
red soils contributed to the genesis of many Terrae calcis, possibly following events
of sudden and considerable pressure such as airbursts during cosmic impacts. Apart
from the presented sand grains, we could not observe pseudosand structures in the
clay matrix of the studied profile, but they could have been lost during shrink-swell
processes. Such processes could also have blurred clay illuviation cutans in the
solum (Fedoroff and Courty, 2013).



One major question is the iron dynamics of the studied soils. The absolute amounts
of dithionite-soluble iron suggest that the Terra rossa crack infillings contain more
pedogenic iron than the Terrae fuscae. This could indicate that iron dynamics are
connected with organic matter contents as suggested by Blanck (1915), since higher
contents of dithionite-soluble iron seem to correspond to lower values of organic
matter. However, there is no correlation to the pH-value, and more important, no
correspondence to the Fe(d/t) ratio. According to Cornell and Schwertmann (2003),
this ratio can be interpreted as indicator of $Fe^{2+}$ oxidation to $Fe^{3+}$, which is usually a
marker of mineral weathering and pedogenesis. Since it remains more or less
constant in nearly all investigated samples, this essentially contradicts the impression
of stronger weathering given by the particle sizes in the lower part of the profiles. The
Fe(d/t) ratio has been problematic in other studies of Terra calcis soil development.
For example, Günster (1999) suggested that it should be modified by the clay content
in order to achieve reasonable results not contradicting other indicators of soil
development intensity, since he observed a strong correlation of $Fe_d$-contents with
clay in southern Spain. In contrast, Lucke (2008) found that the contents of $Fe_d$ in
Terrae rossae of northern Jordan showed some correlation to contents of calcium
carbonate, and suggested modifying the index with the calcium carbonate content.

At the studied profiles in Franconia, however, those two modifications are
inapplicable since there is no apparent connection between the contents of iron,
calcium carbonate, and particle sizes. Since limestone might contain threevalent iron
from sediments, the problematic Fe(d/t) values could be explained by pre-weathered
bedrock residue. However, in light of our microstructural and analytical results

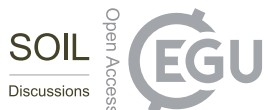

regarding clay neoformation, a different hypothesis is formulated here: that not only
'classical' pedogenesis is connected with the Fe(d/t) ratios in Terrae calcis. Instead,
we regard it possible that the iron content is affected by neoformation of
phyllosilicates involving the transport of external $Fe^{3+}$ ions. If we assume that
aluminium, which is hardly soluble at neutral pH values, can be transported into the
rock-soil transition zone, it seems possible that iron can be transported too, possibly
involving the same transport mechanism. In this context, iron transport by thermal
waters due to volcanism seems possible in the geological context of the
Schwaighauser Forst profile, although there is no direct evidence for that. However,
the Fricke profile seems located rather far from zones of volcanic activity, and there
are so far no clear indications of volcanic ash deposition.

It seems certain that metasomatism and/or illuviation of amorphous clays can
contribute to Terra calcis genesis regardless of the climate or temperature. Further, it
seems likely that the development of red color is connected with strong wetting and
drying cycles as suggested by Cornell and Schwertmann (2003). Related to this, we
think it possible that the Terra rossa of the limestone crack infillings of the Fricke
profile was subject to stronger climatic switches than the upper part of the profile - or
only the upper part was subject to xanthization.

**6.4 Evidence from related studies**
Küfmann (2008) observed that the thickness of Terrae calcis in the northern
calcareous Alps was linearly correlated to the proportion of insoluble residue of



limestone bedrock. However, her calculation of the possible contribution of bedrock
residue (about 20%) and aeolian deposits (about 50%) during the Holocene could
explain only about 70% of the present soil thickness. Longer time periods of soil
formation do not seem probable since the Alps were glaciated during the Pleistocene.
The missing part might be the contribution by isovolumetric replacement and/or
illuviation of amorphous clays, which could also explain the positive correlation of soil
depth with bedrock residue. Since the metasomatic model of Merino and Banerjee
(2008) predicts that acids are produced during the replacement process, the non-
soluble residue of the limestone rocks also contributes to soil development: more
residue will be released in less pure limestones, leading to quicker build-up of the
profile.

For now it has to be left open how the transport mechanism leading to
superconcentration of ions in the rock pores can be explained, but we think that roots
which are present in larger rock fissures and even in the deepest part of the studied
clay-filled cracks seem the most probable transport agents. Verboom et al. (2009)
found that roots of eucalypts colonizing sand dunes in Western Australia were
capable of transporting Al, Fe, and Si, leading to the construction of clay pavements
along the roots in a geochemically alien surrounding. The same elements were found
in the amorphous clays of our studies, which indicates that these probably represent
largely authigenic, newly formed clay minerals that stem from reactions triggered by
plant roots.





Our study suggests that bulk soil and rock analyses alone can deliver only limited
insight into Terrae calcis development, at least at sites where clay neoformation
contributes significantly to the genesis of these soils. It appears that bedrock
weathering does not only proceed by chemical reaction processes which create voids
in the rock due to dissolution, but that neoformation of new minerals and rock
dissolution can be part of the same process. Unfortunately it was not yet possible to
study the mineralogy of the crystalline clays. In the future, a better understanding of
mineral crystallization out of the observed apparently amorphous clays could help to
better explain the factors controlling Terrae calcis formation.

**7. Conclusions**
Our study found amorphous clays in the direct rock-soil transition zones of all studied
profiles. Although it cannot be stated whether these are ongoing or relic features, it
can be concluded that:
- Isovolumetric replacement of limestone due o metasomatism and/or illuviation of
amorphous clays took or takes place in Terrae fuscae as well as Terrae rossae in
Franconia.
- Current topsoil organic matter contents and soil color apparently do not matter for
the occurrence of these features.
- Amorphous clays are observed only close to micropassages in the rock-soil
transition zone, suggesting that rock pore solutions play a role for their occurrence.
These clays do not fill voids, but are present only in contact with calcite structures.



- There is a gradual transition between calcareous minerals and amorphous clay, and
no sharp boundary as would be expected from dissolution and deposition processes.
- The presence of Fe suggests that replacing amorphous clays are either iron-coated
during illuviation, or Fe-ions are transported in a similar way as Al and Si. Since the
same elements can be found in clay pavements around Eucalyptus roots in Western
Australia, this suggests that root activity might play a major role for the formation of
amorphous clays.
- No crystalline illuvial clay could be observed in pores, but allochthonous sand was
deposited into the limestone cracks during loess deposition by wind and bioturbation.
It seems well possible that pseudosand clay aggregates contributed to the solum.
These might explain a part of the substrate, but not the amorphous clays observed in
the rock-soil transition zones.
- The investigated Terrae calcis represent true soils and not claystones, since they
contain a significant share of allochthonous material and are subject to processes
induced by plants – which, by definition, means that pedogenesis takes place.

We conclude that replacement processes can contribute to the genesis of Terrae
calcis in Franconia. It is not yet possible to quantify their contribution, and the
mechanisms of the process cannot yet fully be explained. However, further studies
should consider the possible role of plants for authigenic clay neoformation in Terrae
calcis genesis.





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

Preliminary study on weathering and pedogenesis of carbonate rock. Sci. China Ser.
D. 42(6): 572–581.
Skowronek, A., 2016. Terrae calcis. Handbuch der Bodenkunde, Kap. 3.3.2.9, 1-38.



Stephenson, L. W., 1939. Fossil mollusks preserved as clay replacements near
Pontotoc, Mississippi. J. Paleontol. 13: 96–99.
Temur, S., Orhan, H., and Deli, A., 2009. Geochemistry of the limestone of Mortas
Formation and related Terra Rossa, Seydisehir, Konya, Turkey. Geochem. Intl. 47:
67–93.
Terefe, T., Mariscal-Sancho, I., Peregrina, F., and Espejo, R., 2008. Influence of
heating on various properties of six Mediterranean soils. A laboratory study.
Geoderma 143(3): 273-280.
Torrent, J., Barrón, V., and Liu, Q., 2006. Magnetic enhancement is linked to and
precedes hematite formation in aerobic soil. Geophysical Research Letters, 33(2):
DOI: 10.1029/2005GL024818.
Trappe, M., 2011. Sedimentpetrographie, Gliederung und Genese von
Karstsedimenten, dargestellt am Beispiel der Südlichen Frankenalb. Stuttgart,
Schweizerbart Borntraeger Science Publishers, 195 pp.
TUM, 2014. Technische Universität München, Geomorphologie und Bodenkunde:
Bodenlehrpfad Schwaighauser Forst. http://www.geo.wzw.tum.de/ [31-08-2015].
Verboom, W., Pate, J., and Aspandiar, M., 2009. Neoformation of clay in lateral root
catchments of mallee eucalypts: a chemical perspective. Ann. Bot. 105(1): 23–36.
Weyl, P. K., 1959. Pressure solution and the force of crystallization: a
phenomenological theory. Journal of Geophysical Research 64: 2001–2025.



WRB 2014. World Reference Base for Soil Resources 2014. International soil
classification system for naming soils and creating legends for soil maps. W. S. Res.
Rep. 106. Rome, FAO, 181 p.
Yaalon, D., and Ganor, E., 1973. The influence of dust on soils during the
Quaternary. Soil Sci. 116: 146–155.
Zhu, L., and Li, J., 2002. Metasomatic mechanism of weathering-pedogenesis of
carbonate rocks: I. Mineralogical and micro-textural evidence. Chin. J. Geochem.
21(4): 334–339.
Zech, W., Wilke, B.-M., and Drexler, O., 1979. Analytische Kennzeichnung von
Karstschlotten-Füllungen in der Fräinkischen Alb. Z. Geomorph. N.F., Suppl.-Bd. 33:

922    182-193.

Zippe, W., 1854. Einige geognostische und mineralogische Bemerkungen über den
Höhlenkalkstein des Karst. In Schmidl, A.: Die Grotten und Höhlen von Adelsberg,
Lueg, Planina und Laas. Pleiades Publishing Ltd., Vienna: 209-221.


**Figure and table captions**



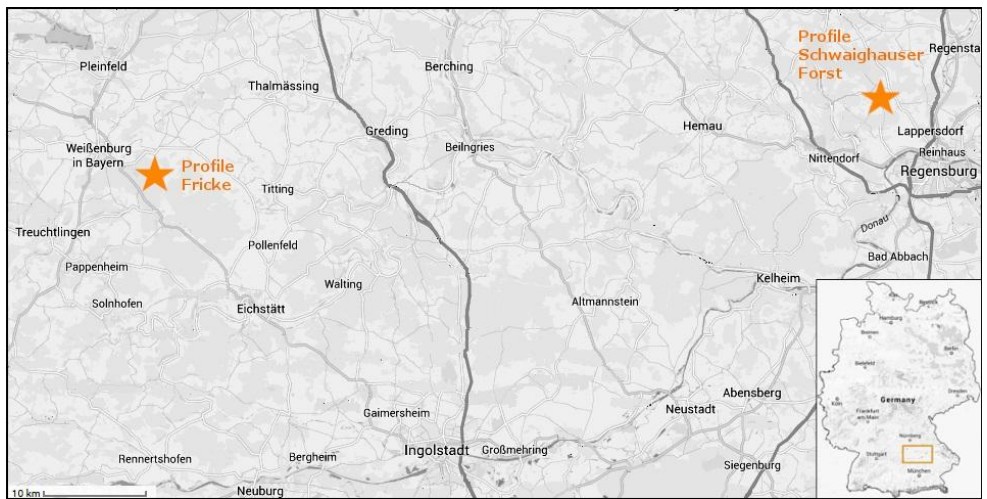


Figure 1: Map showing the location of the investigated profiles in Franconia, and the

region inside Germany. Map based on www.openstreemaps.org (Open Database

License, ©OpenStreetMap contributors).

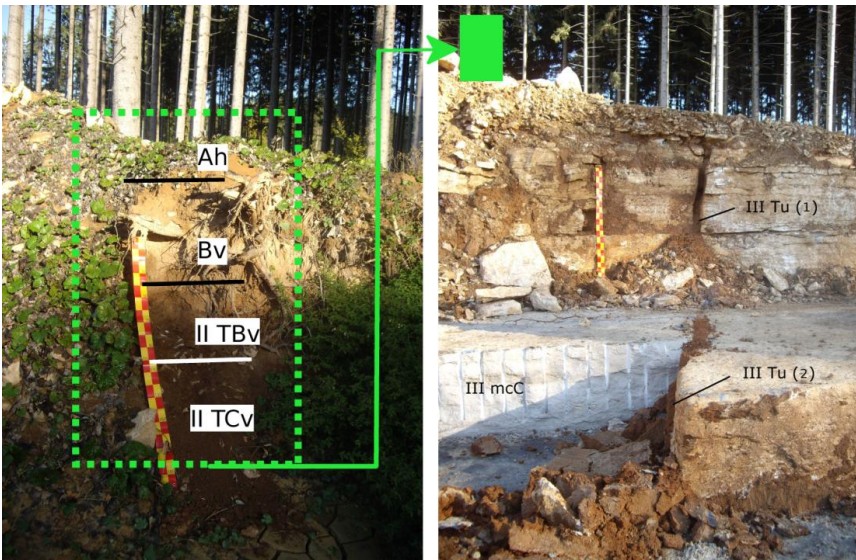

Figure 2a: The profile "Fricke" sampled near Weißenburg i. Bayern. The left side
shows the upper part of the profile, which connects to the lower part shown on the
right side as indicated by the rectangle. Due to a step-wise exposition of rock cuts in



the quarry, it was not possible to obtain a picture showing the whole profile. Sampling
locations are marked by the horizon labels. Each mark on the meter tape represents
10 cm.

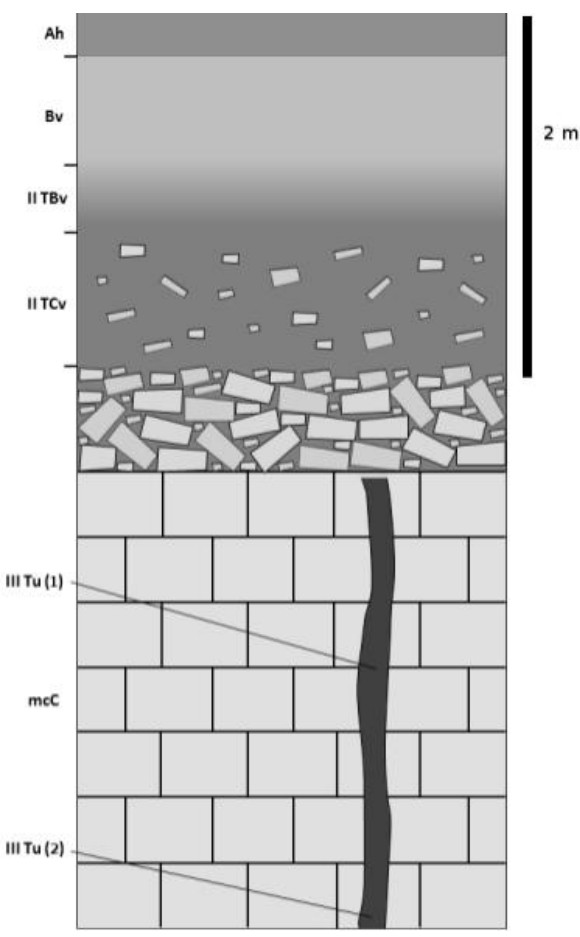


Figure 2b: Schematic drawing of the profile "Fricke" for a better illustration of soil
horizons.




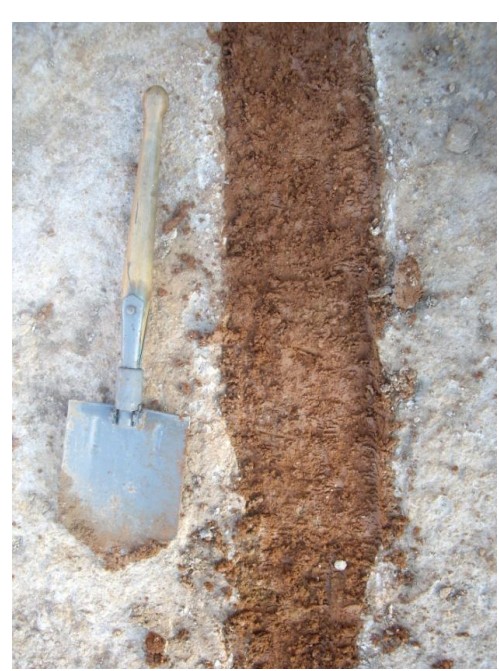



Figure 2c: Close-up of crack infillings. The red clay is homogeneous throughout,
there are no horizons, and no indicators of fluvial deposition could be observed.
Transition to bedrock occurs in a zone of 'powdery' limestone of about 1 mm
thickness as described earlier for other crack fills by Trappe (2011).



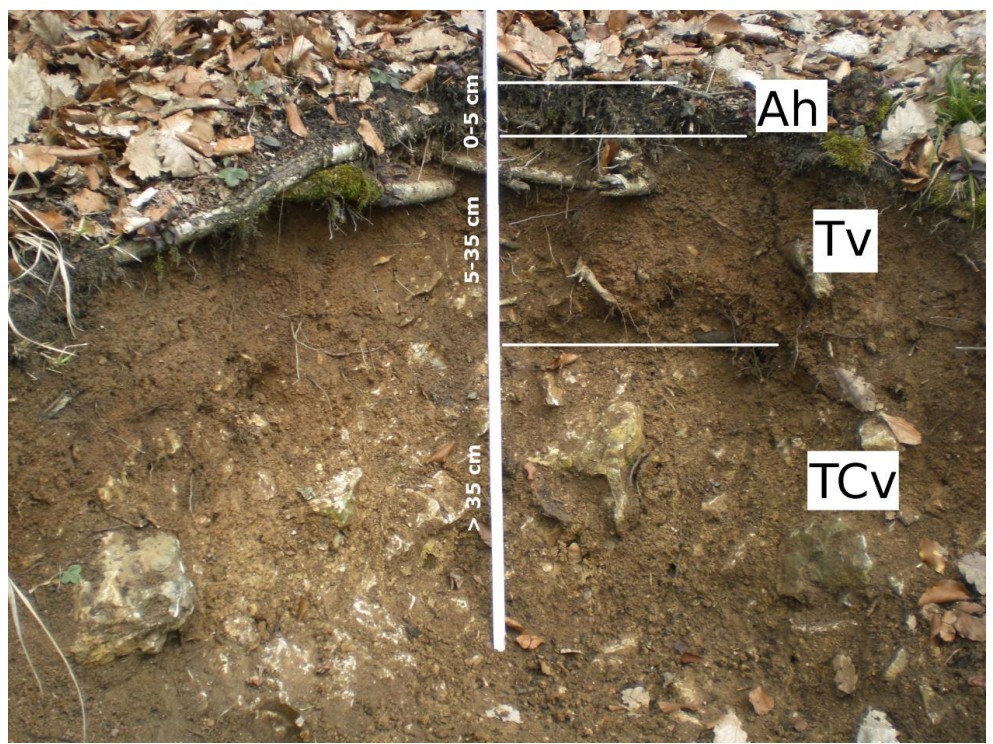


Figure 3a: The profile "Schwaighauser Forst" near Regensburg. Sampling locations
are indicated by the horizon labels, and soil depths are marked along the white bar.

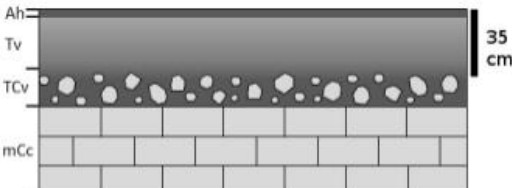


Figure 3b: Schematic drawing of the profile "Fricke" for a better illustration of soil
horizons.




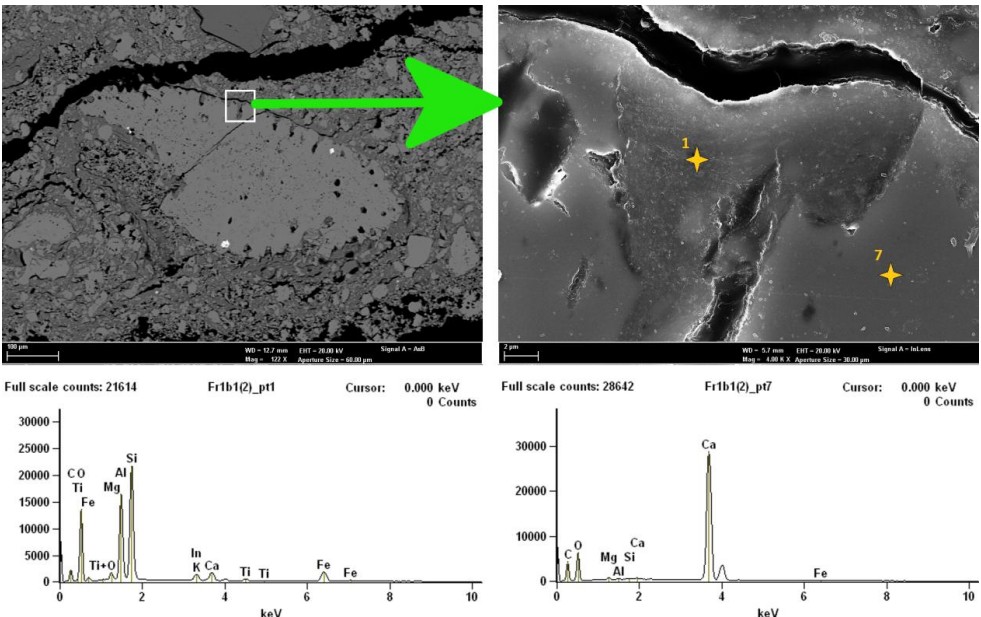

Figure 4: Calcite grain starting to be replaced by clay in the rock-soil transition zone of the Terra Rossa limestone crack fillings in the profile "Fricke". Right is an enlargement of the square marked on the left. The geochemical composition determined by EDS is shown at the bottom for point 1 (left) and point 7 (right), indicating that clay formation maintaining the original bedding structure took place in the darker area of the calcite grain.




972

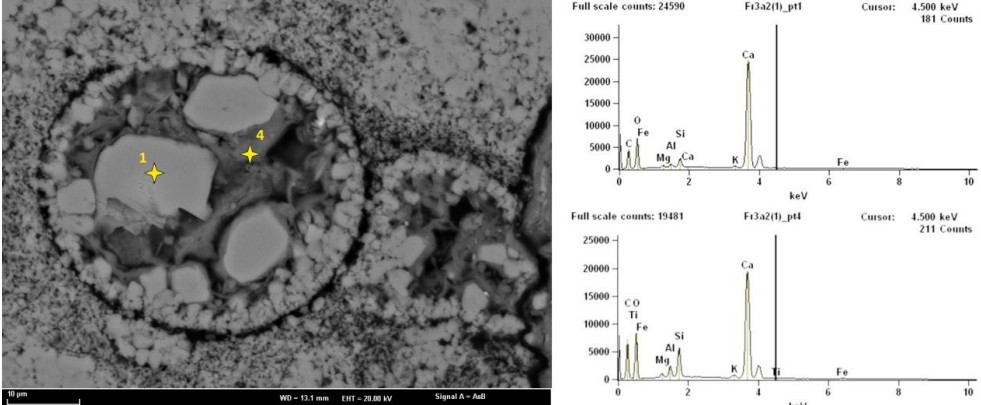

973

Figure 5: Round microfossil partially filled with amorphous clay in the rock-soil transition zone of the Terra Rossa limestone crack fillings in the profile "Fricke". The geochemical composition determined by EDS is shown to the right for point 1 (above) and point 4 (below), indicating that part of the inner microfossil consists of clay although there are no traces of crystalline allochthonous clay or cracks in the fossil's shell.

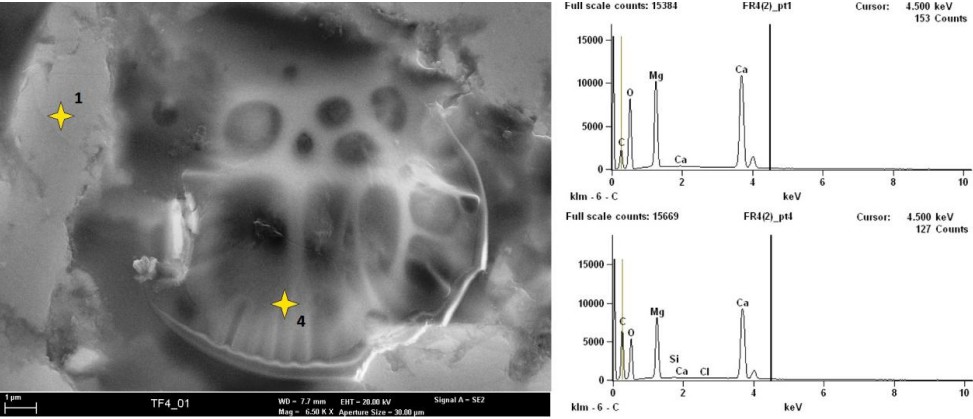





Figure 6: Remains of a microfossil in the rock-soil transition zone of the dolomitic
limestone of the "Schwaighauser Forst" profile. EDS-analyses demonstrate the high
magnesium context of the rock (point 1, top right), but also show a slight increase of
Si and decrease of Ca and Mg in the darker areas inside the microfossil, indicating a
beginning of clay formation (point 4, bottom right).

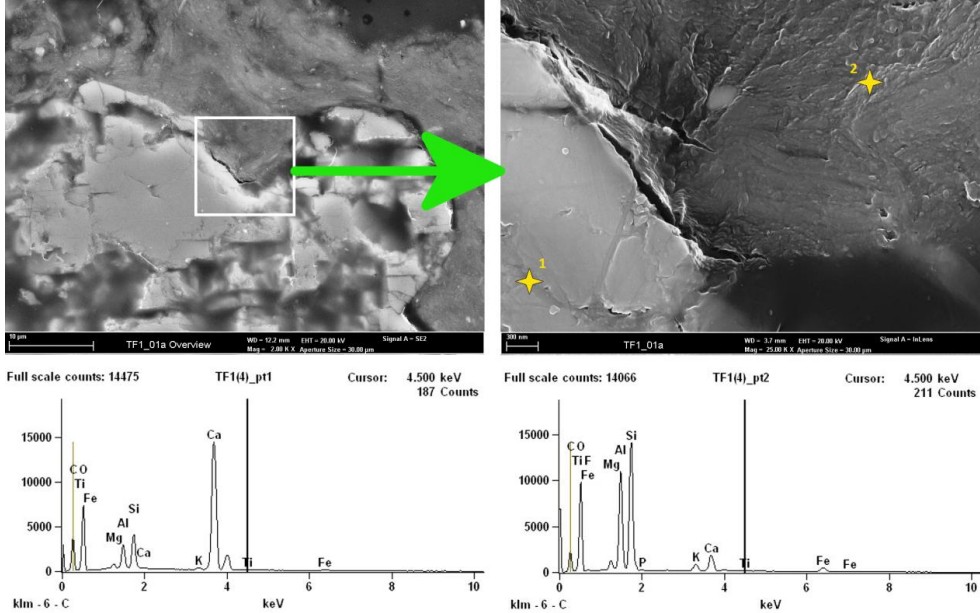


Figure 7: Calcite grain starting to be replaced by clay in the rock-soil transition zone
of the Terra Fusca in the profile "Schwaighauser Forst". Right is an enlargement of
the square marked on the left. The geochemical composition determined by EDS is
shown at the bottom for point 1 (left) and point 2 (right), indicating that clay formation
started inside small cracks of the calcite grain, although chemically still dominated by
Ca (point 1). In contrast, the soil matrix in the larger fissure is characterized by a
further increase of Si, Al, and Fe, while Ca was diminished (point 2).







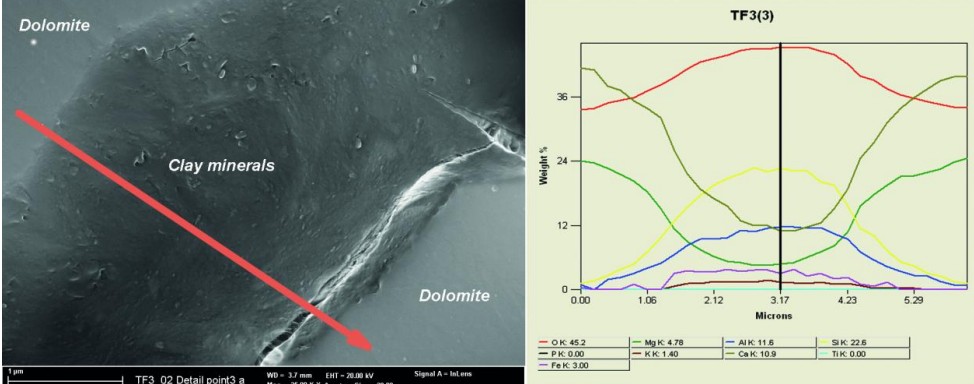


Figure 8: SEM cross-section using EDS to determine element composition along a

section of a dolomite grain in the rock-soil transition zone of the Terra fusca in the

profile "Schwaighauser Forst". The graph to the right shows the weight % of the

studied elements along the section marked by the arrow to the left. As can be seen,

there is a gradual increase of Si, Al, and Fe towards the darker area of the dolomite

grain, while Ca and Mg are reduced, pointing to gradual isovolumetric replacement of

the dolomite by clay minerals.



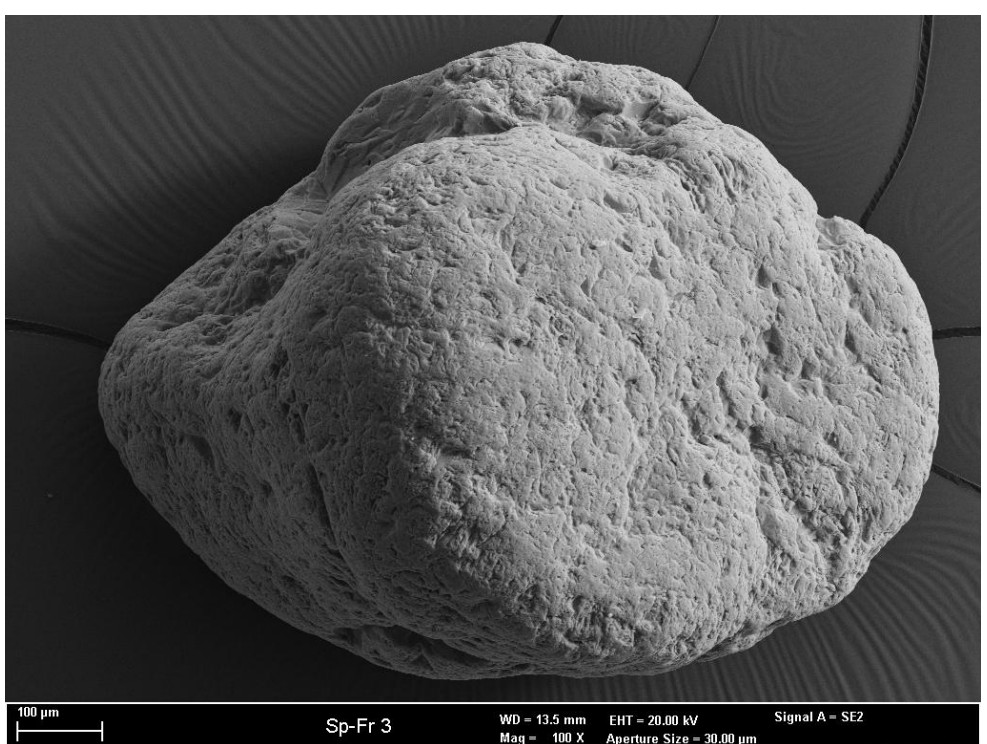


Figure 9: Quartz grain of the coarse sand fraction of the red fill in the limestone crack
of the Fricke profile. Note the densely set small V-shaped marks typical for aeolian
transport during loess deposition.



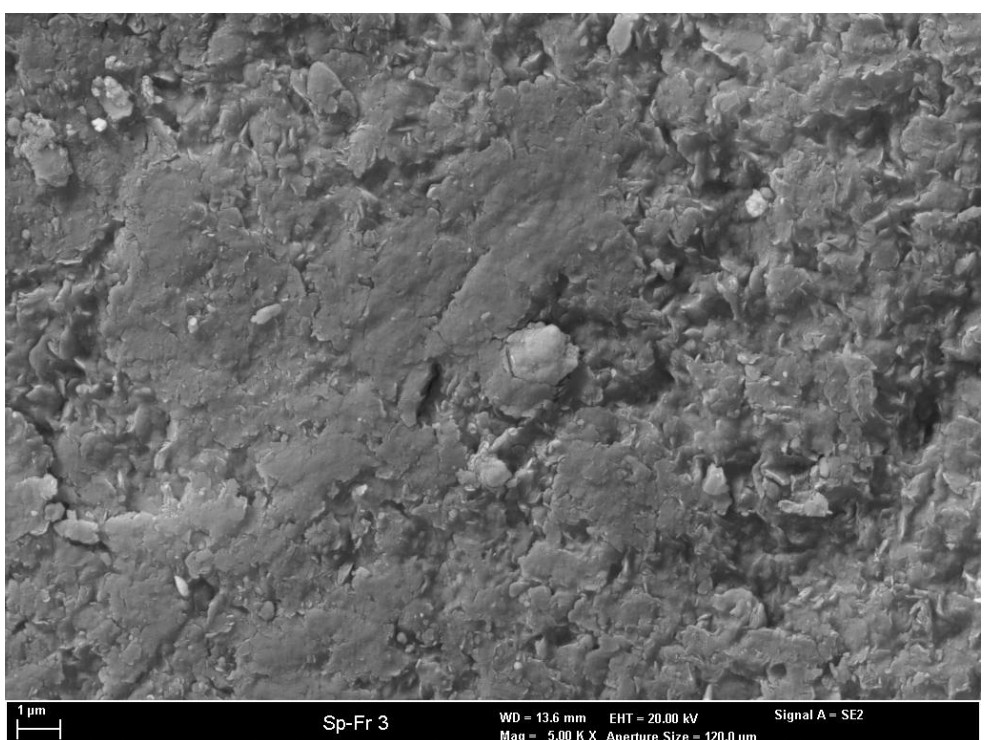


Figure 10: Grain from the fine sand fraction of the red fill in the limestone crack of the
Fricke profile. It is completely covered by clay, possibly representing a pseudosand
aggregate largely consisting of clay (and silt).


Table 1: Results of bulk soil analyses of the profiles "Fricke" and "Schwaighauser
Forst". The Fe (d/t) ratio describes the ratio of dithionite-soluble iron ($Fe_d$) to total iron
contents. Increasing redness is reflected by smaller values in the index according to
Hurst (1977). The dolomite content of the bedrock at the Schwaighauser Forst profile
was calculated according to the residue mass after the rock's dissolution with HCl.



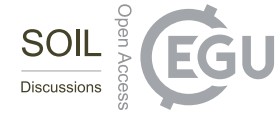

| Sampling depth [cm] | Sample | Munsell dry | RR after Hurst (dry) | Oxalat-soluble silicium [mg/g] | Dithionit-soluble iron [mg/g] | C$_{org}$ % | pH | CaCO$_3$ % | Clay % | Silt % | Sand % | Fe(d/t) | Ti/Zr |
|---|---|---|---|---|---|---|---|---|---|---|---|---|---|
| **Profile FRICKE** | | | | | | | | | | | | | |
| 0-25 | **Ah** | 10 YR 6/3 | 40 | 0.01 | 11.5 | 3.08 | 4.5 | 2.1 | 27 | 69 | 4 | 0.49 | 9 |
| 25-90 | **Bv** | 10 YR 6/4 | 30 | 0.14 | 17.5 | 0.28 | 4.7 | 1.5 | 36 | 61 | 3 | 0.47 | 11 |
| 90-125 | **II TBv** | 10 YR 6/4 | 30 | 0.03 | 19.1 | 0.32 | 5.5 | 0.5 | 45 | 52 | 3 | 0.43 | 12 |
| 125-200 | **II TCv** | 10 YR 6/4 | 30 | 0.07 | 26.6 | 0.35 | 6.7 | 0.6 | 59 | 38 | 3 | 0.54 | 15 |
| 370 | **III Tu (1)** | 5 YR 4/4 | 15 | 0.77 | 33.6 | 0.13 | 7.7 | 2.1 | 65 | 22 | 13 | 0.5 | 18 |
| 520 | **III Tu (2)** | 5 YR 4/4 | 15 | 0.75 | 33.0 | 0.12 | 7.6 | 1.6 | 66 | 21 | 13 | 0.49 | 15 |
| 500 | **mcC** | 10 YR 5/3 | 33 | - | - | - | - | 98.1 | 75 | 24 | 1 | - | 20 |
| **Profile SCHWAIGHAUSER FORST** | | | | | | | | | | | | | |
| 0-5 | **Ah** | 10 YR 6/3 | 40 | 0.02 | 10.1 | 3.80 | 6.5 | 4.6 | 39 | 39 | 22 | 0.38 | 11 |
| 5-35 | **Tv** | 10 YR 6/4 | 30 | 0.04 | 14.3 | 1.10 | 7.6 | 3.2 | 57 | 28 | 15 | 0.43 | 13 |
| >35 | **TCv** | 10 YR 7/4 | 35 | 0.06 | 9.9 | 1.40 | 7.6 | 35.2 | 67 | 23 | 10 | 0.49 | 21 |
| >50 | **mcC** | 10 YR 4/2 | 40 | - | - | - | - | (99.99) | 86 | 13 | 1 | - | 11 |

Table 1: Results of bulk soil analyses of the profiles "Fricke" and "Schwaighauser Forst". The Fe(d/t) ratio describes the ratio of dithionite-soluble iron (Fe$_d$) to total iron contents. Increasing redness is reflected by smaller values in the index according to Hurst (1977). The dolomite content of the bedrock at the Schwaighauser Forst profile was calculated according to the residue mass after the rock's dissolution with HCl.