# Peer review of "Isovolumetric replacement and aeolian deposition contributed to Terra calcis"

_SOIL, 2017_

## Referee Comment (RC1) · Anonymous Referee #1 · 28 Feb 2017

The paper entitled "Isovolumetric replacement and aeolian deposition contributed to Terrae calcis genesis in Franconia (central Germany)" is an article that supplies to the debate about the origin of Terra Rossa and its soil formation processes. This research support the theory wide discussed since the first publication of Merino and Banerjee (2008) about an allochthonous material and its weathering that contributes to replace carbonates in the bedrock to generate soil. The article is well written in English, however, many aspects of the technical vocabulary associated to mineralogy, petrography and soil micromorphology are confusing generating speculation (this will be mentioned in the corresponding section). With respect the Introduction, in the section 3.1 (Meta-somatism in Terrae calcis genesis), a more profound revision of the concept of meta-

somatism is necessary, for example, considering the publications of Putnis (2002), Putnis & Austrheim (2010) and Kondratiuk et al. (2015). In the section two, Genesis of Terra calcis in Franconia (central Germany), is notable the descriptions of the "crack fills". Many of these soil infillings have been described as pedosediments, pitfalls, even palaeokarst, in tropical environments of Yucatan, Bahamas and Bermuda (e.g. Carew & Mylroie, 1991);at respect, the figure 2, in particular 2a and 2c generates questions about its morphology. The description denotes the homogeneous aspect of the infilling and very abrupt and vertical limits of crack, even its continuity to the deepest part of the profile...could be possible to suggest the presence of a fault plane and fault gouge material? I think is necessary more detail in the geology description of the study sites, including lithology and major structures must be showed in the location map (figure 1). The map must be developed under cartography qualities. The section fourth (methods) denotes the most controversial sections of the paper, unfortunately this affect the results and its discussion: the observation scale. The authors state that micromorphology was applied to observe the clay neoformation and its relation with carbonate bedrock. That's inexact, in any case what was used were ultramicroscopy techniques. The preview step to use ultramicroscopy is a medium scale study with thin sections. The authors, apparently, used a thin section to apply EDS, however, they never make reference to the general description of the samples under the petrographic microscope to localize a potential area with isovolumetric replacement. The authors base much of their work in the propose of Merino and Banerjee (2008) which defend the use of petrography descriptions to identify isovolumetric features. Additional to this omission, is the decision to not made petrography of the "unaltered" limestone and describe its components. This aspect is the key to compare with the "altered" sections in the infillings. Another aspect necessary for revision is the decision to concentrate the study of isovolumetric replacement in the microfossils; is well know the different processes associated to authigenic clay minerals in carbonates in diverse sedimentary deposits (e.g. Bristow et al., 2012; Deocampo, 2015; Ifandis et al., 2015). The authors never contrast information about these possibilities in the discussion section. The general

impression is that the authors pretend to see the same evidence and processes that Merino and Banerjee (2008) in Indiana, biasing and even forcing their results. The section five, the results, are confusing in the descriptions, beginning with the "micro-morphology section", whit descriptions as "clay films", "allocthonous clays", "micro-clay beddings" (particularly in the manuscript lines 385-391), vocabulary used without context and not used in the technical terminology of micromorphology. Part of the results is more close to a discussion section, even "conclusions" as shown the lines 391-394. Specially, expressions as "amorphous structure" or "amorphous gel" generates several confusions in the reader: do you mean "low crystallinity" or "low birrefringence" (if you really observe thin sections under petrographic microscope)? I don't know if try to describe "clays aggregates" or "clay coatings with low birrefringence" or even "isotropic clay coatings". The paper needs an important revision of vocabulary concerning to soil micromorphology and carbonates petrography. In the section 5.2 (bulk soil analysis), again the paper present a mix of results and discussion; I think is recommendable to prepare just one section with the title of "Results and Discussion" . Is necessary to revise the assertion that v-shape marks on the quartz grains are typical of wind transport (please, check publications as Reineck and Sing (1986) and the recent work of Woronko and Pisarska-JamroÅijy (2016) associated to frost weathering); if you work with "real" micromorphology, probably you can observe some evidences of the sedimentary processes related with the input or transport of the "infilling" in the cracks. A category of Mineralogical results is essential, contrasting the mineral phases in lime-free residue, cracks infillings and soils. The mineralogical interpretation is vague and poor understood; the same is with the major and trace elements, never used in the paper to interpret processes, just a part with the EDS results and Ti/Zr ratios (it is obvious the dissolution effect of these elements in the limestone and its concentration in the soil and infillings). An important question is to know which mineral phases are concentrated in these elements ( zircon, rutile, epidote group minerals, others? ) I think that much of the discussion concerning to Parent Materials (6.1) must be related to mineralogy and geochemistry. The Discussion section denotes weakness in the understood of metasomatism concept background. Again, the major question is : do you really have concrete evidence of neoformed clay minerals beyond photomicrographs by SEM and the use of explanations as "clay gel like-material"? Descriptions as "no crystallinity" and/or "amorphous clays" are very ambiguous in the text and scarce informative and discussed. I recommend consulting the publication of Velde and Meunier (2008). The petrographic aspects of the carbonates, specially related to needle calcite forms can be discussed with the help of publications as Verrechia (2011) and his review of pedogenic carbonates. With respect the bulk soil analyses, there is a poor exploitation of the geochemical data (major and traces elements); the paper discuss Ti/Zr ratios without references to the possibility to use elements for provenance studies by comparison with other published data. The loess influence must be traced with the help of REE patterns and the use of Y, Th, La, Zr among others elements in ternary diagrams. Aspects as the speculative presence of "pseudosand structures" and the "shrink-swell" in the soils must be verified by micromorphology in the soil groundmass and its b-fabric.

---

## Author Comment (AC1) · 14 Mar 2017

Dear reviewer #1, dear colleagues,

thanks very much for your efforts invested in our paper. We agree that some of the terminology should be improved, and that it is a good idea to include the actual results and discussion in a joint chapter "Results and discussion". However, some recommendations seem a little out of the scope of this article. We do neither intend to present a complete review of the literature on Terra rossa genesis (which would require many more pages describing the state of the art), nor a full-scale petrographic investigation of the studied profiles. As well, we do not intend to quantify the contribution of possible individual parent materials such as loess, replacement, or bedrock dissolution, and

think this has been made clear in the introduction.

The scope of this paper is intentionally limited to the qualitative reporting and discussion of features in the rock-soil transition zone that might be explained by isovolumetric replacement, and to some pedogenic parameters, in particular the sand content, that might be attributed to deposition of allochthonous material. The cited literature has been limited to the sources that we assume essential for offering a thorough interpretation in the context of the current state of the art. We consider this legitimate since the vast literature on Terra rossa genesis is more or less impossible to summarize in a paper presenting original data – a selection has to be made unless one aims to write a review paper. For example, sources like the mentioned papers by e.g. Putnings and Kondriatuk et al. are very specific. They focus on models of linear kinetics that could provide an analytical expression of chemical mechanisms by which a tight coupling between precipitation and dissolution fronts can arise that lead to volume-preserving replacement. They can be cited, but we consider that not necessary since these contributions deal with very specific details of the possible replacement process that might not really matter for our argumentation.

Similarly, it might currently be impossible to calculate the contributions of various possible parent materials from elemental compositions: since we do not know the extent and origin of possibly neo-formed clay minerals, such statistical estimates must remain to some degree speculative, distract from the qualitative reporting of the observed features, and would greatly extend the length of the paper. This intentional limitation explains our 'poor' use of the available geochemical data. Similarly, we do not agree that the mineralogical analysis of the studied profiles is be needed within our scope. That comparisons of clay mineralogy are of limited use has been showed by recent studies by Sandler et al. (2015) who found that weathering can be reversible and similar clay mineralogies are thus not suited to prove inheritance. Regarding the sand contents, it would certainly be of interest to better understand the mineralogy of the grains for tracing their origin, but already the available data allow for sufficient conclusions in the
context of the scope of our paper.

It is possible to provide better maps and microphotographs of the limestone. However, we are not certain whether this matters for the reported evidence. As described in detail in the paper, the possible replacement features were only visible at the rock-soil transition zone, while the limestones consisted of calcite or dolomite with microfossil inclusions. Such a microphotograph might be presented, but then as evidence that there is nothing new to see, and we are not sure how important that would be. In addition, areas of unaltered limestone that border the observed replacement features are already visible in the provided microphotographs.

What is true is that the paper presents only results of ultramicromorphology – a method never applied by Merino and colleagues. As explained above, we focused on the possible replacement features which could only be observed by SEM and EDS and did not present results of micromorphology as the latter provided no new information. We therefore agree that some revision of the terminology and description is indicated. However, the review provides no recommendations on the observed key features, and we consider the question "do you really have concrete evidence of neoformed clay minerals beyond photomicrographs by SEM" as not fully fair. Photomicrographs are the only possible evidence... the difference between the features reported in our paper and the ones observed by Merino and Banerjee (2008) is the use of SEM, and even though the features that we observed might not fully match the theoretical model proposed by Merino and Banerjee (2008), we see no reason to discard them just because they could not be observed by optical micromorphology.

Unfortunately the review does not provide an alternative explanation for the observed features, but is in a way discussing them away by demanding more methods, quotations, and additional data. But the observed features are very similar to the ones observed by Lucke et al. (2012), although that study dealt with soil development in semi-arid Mediterranean climate. One motivation to conduct our research was to check whether similar features can be observed in temperate climates, and we consider the

fact that this is the case as already highly relevant for various questions of soil science even though some aspects cannot yet be explained or quantified.

Regarding the suggested map of geological faults and fractures, it is well-possible that the studied limestone cracks are connected to regional fracture systems. But again, the question is whether this matters for the studied clay fill and its rock-soil transition zone. While it is certainly possible that transport of allochthonous material might have proceeded via regional fracture systems, it would be out of the scope of our paper to attempt tracing such movements. Even with a geological map, we cannot prove or disprove whether fault plane and fault gouge material is present. We think not, since there is a strong similarity to the covering lower Terra rossa material and sand grains of possible glacial loess origin, which will be added in the discussion. But even if fault plane material is present, that would not change anything since the questions of its origin and formation remain.

What can be stated is that an increase of sand content by 1300% compared to the bedrock residue, and by 300% compared to the covering loess, is very likely connected with the deposition of allochthonous material, and v-marks on the quartz grains are a strong hint to a glacial origin of such grains (frost weathering would even more associate them with the loess, since aeolian transport might also have occurred in the reef environment where the limestones formed). A standard reference is Mahaney (2002): aeolian transport of the grains is unambigious, and a connection with loess deposition very likely – although, as stated in our discussion, that is only one possible source of the sand and the question of transport mechanism into the cracks remains.

In this context, grains that appear as sand size during bulk soil analysis, which show under the microscope to be hard rounded aggregates of finer particles, are by definition 'pseudosand' – as well as macroscopic prismatic aggregate structures in clay-rich soils are known to result from shrink-swell processes. We are thus not 'speculating' about pseudosand. It is there, and part of the key question how to explain the increase of the sand content compared to the bedrock residue.

With regard to the key evidence of our paper, i.e. the features partially consisting of clay and calcite observed by ultramicromorphology, and regarding the strong increase of sand content, the review did in our opinion not provide substantial suggestions. We therefore agree to improve the terminology and description and unite the results and discussion in one chapter, but would prefer to leave the scope and extent of the paper as it is.

Thanks and best regards Bernhard Lucke, Helga Kemnitz, and Stephan Vitzethum
* * *

---

## Referee Comment (RC2) · Anonymous Referee #2 · 22 Jun 2017

I went thoroughly through this manuscript and had to conclude that its quality is poor to the extent that I have no interest to spend any further effort in its improvement or so. The authors seemingly do not know what they are writing about and have no idea of the way in which up to date studies on such residual soils should be performed. It is just a small number of EDS-data on a few thin sections - the only somewhat more 'novel' data - and these results are stretched and interpreted in a way which I would never have accepted from one of my MSc students. Moreover, it blunders through the complexities of the geochemistry of such systems and is full of unfounded statements and assumptions. Could it be improved: yes but only if at least 2/3 of the manuscript concerned with all these unfounded hypotheses, interpretations and conclusions are

eradicated (and not to forget, completely irrelevant references) and it would be based only on true results and their unbiased interpretation. I am not inclined to aid them in that, since I am afraid that they simply miss the required expertise and attitude. The latter has to do with the style of the manuscript: in my long career I have seen few such vague, biased and hypothetical papers with so little data.

Please also note the supplement to this comment:
http://www.soil-discuss.net/soil-2017-4/soil-2017-4-RC2-supplement.pdf

———————————————————————

[Figure]

**Supplement:**

One of the major problems with this manuscript is that it concerns a topic which has already been extensively studied by many soil scientists. Approaches to be used in this kind of studies on residual materials and analytical methods are well established. Crucial questions include:

- can the composition (both chemical and mineralogical) of the non-carbonate components of the 'residual' material be linked to the non-carbonate fraction of the limestone;
- if a residual origin must be assumed, how much limestone should have been dissolved and what implications does this have for the geomorphological development of the area.
- What may be the origin of the materials that do not originate from the original limestone, and that goes far beyond the ubiquitous loess cover, but today includes such materials as volcanic (micro)tephra, Sahalian dust, etc.
- To what extent are materials encountered the result of inheritance of early depositional characteristics or diagenetic processes (e.g. chert formation) or the result of much more recent subaerial weathering.
- In case of assumed recent neoformation: can the flux required for the production of the presumedly neoformed materials (stocks) be accounted for/ be explained. This requires a good understanding of the geochemistry of the system and fluxes of solutes that are possible under the specific conditions.

Crucial techniques include:

- Full chemical and mineralogical analyses of the soil material and the limestone and its residue, including a proper identification of the clay fraction. The latter has to include XR-diffraction data and not only EDS or other chemical analytical techniques.
- A proper description of the micromorphology of the material to have a good understanding of its fabric and provenance of components (e.g. clay, nodules, concretions, etc.).
- Eventually isotopic analyses to identify the origin of the various minerals encountered (e.g. Sr and Nd isotopes to establish the origin of tephra).
- If current soil formation and weathering are to be included in the research: information on the chemical composition of the soil solution and its link to geochemical processes.

The current manuscript basically consists of "static" chemical data at the nanometer scale (EDS) providing only information on element distribution in the soil matrix, and some general data on the distribution and habitus of Fe species (apart from standard very general soil data). That is all.

- there is no information on the chemical/mineralogical composition of the non-carbonate components in the residual soil nor in the limestone
- there is no full study of the potential provenance of the non-carbonate fraction
- there is no information on the presence of original or diagenetic features in the limestone (for example the presence of clay in fossils, which may well be of very early age and the presence of silica)
- there is no understanding at all of the geochemistry of these systems and behavior of the various species in this system.

As to techniques:

- no full chemical and mineralogical analyses, particularly no XR-diffraction data
- no full study of the potential provenance of the non-carbonate fraction.
- no micromorphology
- no isotopic analyses nor full study of trace elements to study the potential provenance of the various materials/components. Typical example is the Zr/Ti ratio which clearly indicates that the residual fill has very little to do with a dissolution residue from the limestone.
- no understanding of the geochemistry of these systems, e.g. the assumption that Al3+ plays a role in these systems and occurs as a solute, leading to isovolumetric substitution. Study of current composition of the soil solution and speciation of the solutes would easily have shown that.

Moreover, all kinds of terms are being used in connection with the apparently major phenomena: occurrence of clay size material in small voids: Isovolumetric replacement, pressure-driven metasomatic replacement, authigenic clay neoformation, exchange process characterized by substitution, pressure-driven isovolumetric replacement during authigenic clay neoformation, replacement processes, metasomatic processes.

There are very fundamental differences between the various processes basically coming to: a) precipitation from the solution and b) mass transport as suspended material in the solution. 'Replacement processes' is a meaningless term in this context. This holds the more, while 'replacement' cannot be observed as an active process, but is merely an interpretation of spatial structures and distribution of elements. What is dearly needed is a clear definition of the various terms and their strict application + arguments that exist for one or another interpretation. Now it is a mess and often completely obscure what is truly meant.

All in all, I am not very happy about this study, which in fact has only a few EDS results as 'new data' and rambles on from one assumption to another, and from quite poorly founded conclusions to bland nonsense. Moreover, the very extensive literature review is incomplete, missing major and highly relevant studies, highly unbalanced (a lot of completely irrelevant studies), and poorly structured, mixing results with assumptions and conclusions, and not to the point.

Typical example of the quality is the text in the lines 737-741: Extremely vague: can contribute – not yet possible – cannot yet be fully explained – further studies - should consider the possible role. In other words, one asks oneself what the contribution of this paper is if these are the conclusions. Not a very impressive contribution to science and that is also my general conclusion: *not a significant contribution to science at all. Just a few observations with EDS on some thin sections from residual limestone soils. Much too far reaching interpretations and too many unsubstantiated claims based on these limited data.*

[revised manuscript text omitted]

---

## Author Comment (AC2) · 23 Jun 2017

Dear editor, reviewer 2,

thank you for your efforts invested in our study. We found review #2 quite irritating, especially since the reviewer felt obliged to descend to a language level close to the sphere of personal insults. Unfortunately, when this sphere is reached, one often maneuvers in the realm of battles of faiths and we feel it is not advisable to waste time on this.

To keep it short, we do not agree with this review. It demands extensive data on clay

mineralogy from X-ray diffraction and on the geochemistries of rock and soil, which are however not suited to understand features of the very thin rock-soil transition zone. It is impossible to carry out bulk soil analyses of such small areas. In addition, such bulk analyses are based on outdated concepts, methods, and models of Terrae calcis research. There is no point of conducting a comparison of the clay mineral assemblages of soil and bedrock residue any more, since it has been shown that e.g. illite neoformation during pedogenesis is possible (see e.g. Sandler et al. 2015, possibly due to a catalyzing role of plants). Similary, there are serious doubts whether is it possible conduct a comparison of the geochemistry of clays with the bedrock (residue) since even weak acids might induce considerably stronger leaching (including various rare elements) than previously assumed. That such comparisons were conducted extensively in the past does not help, and it is not useful to cite all these studies. In this context, one could certainly have studied the current composition of the soil solutes, but this would not help to address the potential role of plant roots, or the question whether replacement features are relict.

The only way to investigate the geochemistry and structure of the thin rock-soil transition zone are "static" chemical data at the nanometer scale as in our study, since even classical petrography with optical microscopes is not suited for this scale. However, it should be pointed out again that the reported features could be observed only in the rock-soil transition zone, and not in the soil matrix or the rocks. This restriction to the transition zone unfortunately seems to have escaped notice of reviewer #2. In this context, we find it disappointing that the observed features were not discussed in the review, but flatly discarded as "just a few observations with EDS on some thin sections". It should be pointed out that they could be observed in practically all thin sections of the (more numerous) rock-soil transition zones that we studied, so they are a quite common phenomenon, but one has to limit the number of examples in a publication. Unfortunately, the reviewer does not dare to explain why spatial observations at nanoscale should be so completely useless. We could imagine sedimentary features of the primary limestone as alternative explanation, and it would have been helpful if

respective sources with similar observations would have been provided. Yet the question would remain why no such features could not be detected in the limestones of our profiles, but only in the narrow rock-soil transition zones.

Although it is difficult to imagine phyllosilicate or Al-mobility in a limestone system, we do not consider it good scientific practice to disregard the growing literature on replacement as "a lot of completely irrelevant studies". We have the impression that reviewer #2 was either not willing or not competent to provide a balanced evaluation of our observations. Whether finally ending in the soil discussion forum or a soil paper, this public review has at least the advantage that our observations and their (rather close-minded) reception are now publicly available and not suppressed. Disappointingly, we feel that nanoscales studies of potential replacement features are not welcome to reviewer #2, perhaps because they could place question marks behind the conclusions of various of the above mentioned earlier studies.